

# Interrater reliability of quantitative ultrasound using force feedback among examiners with varied levels of experience

Michael O. Harris-Love[1,2,3], Catheeja Ismail[1,3], Reza Monfaredi[4], Haniel J. Hernandez[1], Donte Pennington[1], Paula Woletz[1,5], Valerie McIntosh[1,6], Bernadette Adams[1,6] and Marc R. Blackman[6,7,8]

[1] Muscle Morphology, Mechanics and Performance Laboratory, Clinical Research Center, Veterans Affairs Medical Center, Washington, D.C., United States
[2] Geriatrics and Extended Care Service, Veterans Affairs Medical Center, Washington, D.C., United States
[3] Department of Exercise and Nutrition Sciences, Milken Institute School of Public Health, George Washington University, Washington, D.C., United States
[4] Sheikh Zayed Institute for Pediatric Surgical Innovation, Children's National Hospital, Washington, D.C., United States
[5] Health Sciences Division, Howard Community College, Columbia, MD, United States
[6] Research Service, Veterans Affairs Medical Center, Washington, D.C., United States
[7] Departments of Medicine, Biochemistry & Molecular Medicine, School of Medicine and Health Sciences, George Washington University, Washington, D.C., United States
[8] Departments of Medicine and Rehabilitation Medicine, School of Medicine, Georgetown University, Washington, D.C., United States

Corresponding author
Michael O. Harris-Love, mhl@gwu.edu, michael.harris-love@va.gov

## ABSTRACT

**Background.** Quantitative ultrasound measures are influenced by multiple external factors including examiner scanning force. Force feedback may foster the acquisition of reliable morphometry measures under a variety of scanning conditions. The purpose of this study was to determine the reliability of force-feedback image acquisition and morphometry over a range of examiner-generated forces using a muscle tissue-mimicking ultrasound phantom.

**Methods.** Sixty material thickness measures were acquired from a muscle tissue mimicking phantom using B-mode ultrasound scanning by six examiners with varied experience levels (i.e., experienced, intermediate, and novice). Estimates of interrater reliability and measurement error with force feedback scanning were determined for the examiners. In addition, criterion-based reliability was determined using material deformation values across a range of examiner scanning forces (1–10 Newtons) via automated and manually acquired image capture methods using force feedback.

**Results.** All examiners demonstrated acceptable interrater reliability (intraclass correlation coefficient, ICC = .98, $p < .001$) for material thickness measures obtained using force feedback. Individual examiners exhibited acceptable reliability with the criterion-based reference measures (ICC > .90, $p < .001$), independent of their level of experience. The measurement error among all examiners was 1.5%–2.9% across all applied stress conditions.

**Conclusion.** Manual image capture with force feedback may aid the reliability of morphometry measures across a range of examiner scanning forces, and allow for consistent performance among examiners with differing levels of experience.

Rehabilitative ultrasound imaging (RUSI) is an approach to diagnostic sonography that incorporates both quantitative and qualitative assessment techniques to characterize musculoskeletal tissue and aid the implementation of therapeutic interventions (*Harris-Love et al.*, *2014*). RUSI applications are typically used to quantify post-intervention changes in tissue morphology, obtain joint space measures, provide visual biofeedback during therapeutic exercise, and further elucidate the contributions of muscle structure and neuromuscular activity to physical performance (*Teyhen*, *2007*; *Blazevich et al.*, *2007*; *Whittaker & Stokes*, *2011*).

Musculoskeletal assessment involving RUSI may feature quantitative techniques that differ from other uses of sonography (*Harris-Love et al.*, *2016*). Quantitative ultrasound imaging techniques are emerging as a non-invasive approach for describing muscle morphology in patients with neuromuscular disease and age-related dysfunction (*Janssen et al.*, *2014*; *Ismail et al.*, *2015*). However, this application of ultrasound is dependent on a specific set of examiner psychomotor skills, such as force, which may affect key measures of tissue dimensions and image echogenicity. Other investigators have demonstrated that variations in examiner scanning force may yield errors in the measurement of muscle tissue thickness (*Ishida & Watanabe*, *2012*; *Harris-Love et al.*, *2014*). These previous investigations describing the impact of examiner performance on ultrasound image acquisition and quantitative assessment have involved both human subjects and ultrasound phantoms.

The concept that both normal and pathologic tissue can be simulated, thus allowing learners to acquire procedural skills while minimizing patient burden, has resulted in the development of an array of ultrasound phantoms and simulators. While the initial use of phantoms in sonography was driven by the need to calibrate ultrasound devices (*Woo*, *2002*), tissue-mimicking ultrasound phantoms are now frequently used to train clinicians. Practitioner training experiences featuring tissue-mimicking ultrasound phantoms may be used to instruct ultrasound-guided invasive procedures or assist investigators in the development and validation of new ultrasound applications.

Given the potential examiner dependency associated with quantitative ultrasound techniques, we propose that the use of force-feedback scanning will foster the acquisition of reliable morphometry measures under a variety of scanning conditions. In this study, we determine the reliability of force-feedback image acquisition and morphometry over a range of examiner-generated forces using a muscle tissue-mimicking ultrasound phantom. We obtained reliability estimates for feedback-enhanced sonography using two methodological approaches. First, interrater reliability among the six examiners was determined based on material thickness measures obtained using manual force-feedback scanning and a series of applied force targets. Second, the criterion-based reliability of material thickness measures was determined by comparing the values obtained from each examiner using manual force-feedback scanning with the values obtained with robot-assisted force feedback scanning.

We hypothesized that the manual force-feedback image acquisition method would yield reliable morphometry measures among the examiners, and in comparison to the criterion values obtained using robot-assisted image acquisition. In addition, we posited that the examiners would exhibit similar criterion-based reliability for morphometry measures, independent of experience level.

## MATERIALS AND METHODS

### Ultrasound phantom

The scanned material was a custom muscle tissue-mimicking ultrasound phantom (i.e., anechoic gel, 15 kPa; speed of sound, 1,540 m/s; attenuation, 0.1 dB/cm/MHz; CIRS, Inc.). Water-soluble transmission gel was used during scanning to attain optimal acoustic contact with the imaging site. Preliminary images were initially obtained in the transverse and longitudinal view to orient the examiners to the ultrasound phantom and to aid the calibration of the force-feedback transducer interface system. Longitudinal view image capture was completed in both the automated and manual scanning conditions at the midpoint of the ultrasound phantom for data collection purposes.

### Examiners

The study was approved by the Washington DC VAMC's Institutional Review Board and Research and Development Committee (IRB; #01671). Scanning was completed by six examiners who were categorized based on their level of sonography experience. The two "experienced" examiners had each used diagnostic ultrasound in clinical or research settings for over a decade. One examiner, a licensed physical therapist (M.H.L.), had previous experience with quantitative musculoskeletal ultrasound, and the other examiner (P.W.) had primary experience as a registered sonographer in hospital and outpatient settings. The two examiners categorized as "intermediate" were health professionals, one a licensed physical therapist (H.J.H) and the other a registered nurse (B.A.) with approximately one year of quantitative ultrasound experience in a research setting. The "novice" examiners were research assistants with approximately one month of quantitative ultrasound experience. One of the novice examiners had a background in exercise science (V.M.), while the second novice examiner had a background in biomedical engineering with experience in a hospital-based laboratory involved in clinical studies (R.M.). An experienced clinician (C.I.) with over two decades of clinical sonography experience provided all of the examiners with basic instruction in the operation of the ultrasound machine, image capture using the ultrasound phantom, digital morphometry measures, and use of the force-feedback interface system.

### Force feedback ultrasound materials and approach

All images used in this study were obtained via B-mode scanning using a portable ultrasound machine (SonoSite M-Turbo 1.1.2; SonoSite, Inc., Bothell, WA, USA) with a 13.6 MHz linear array transducer. The ultrasound machine was operated using its default gain levels and the "musculoskeletal" scanning factory preset. The transducer was fitted with a custom interface housing designed with SolidWorks software (version 2014 x64; Dassault Systèmes

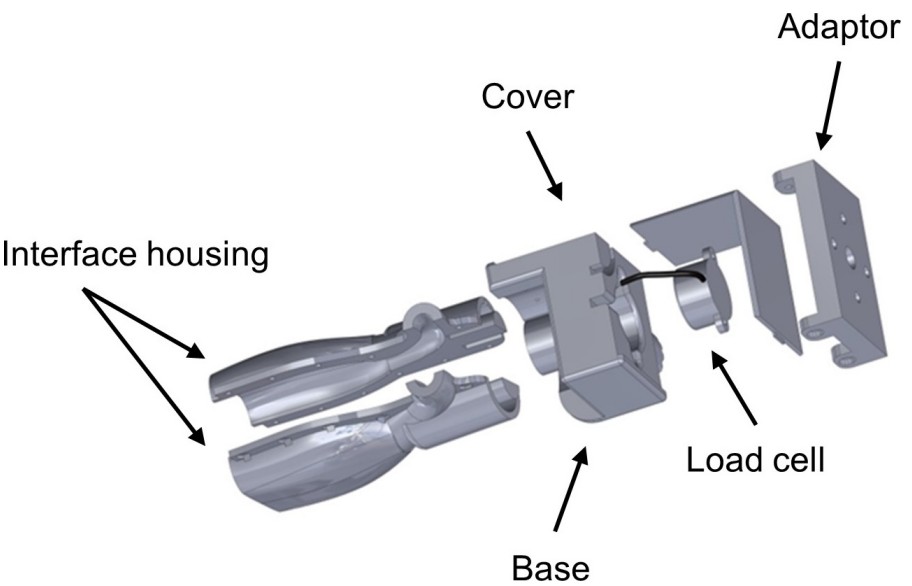

**Figure 1  Ultrasound transducer interface housing for the load cell used during force-feedback scanning.** The figure depicts an exploded view of the custom ultrasound transducer interface housing that was used to connect the load cell to the ultrasound device in order to detect examiner forces without impeding scanning. The augmented ultrasound transducer was used manually for hand-held image capture and also attached to the KUKA Light Weight Robot end effector for use during automated image capture.

SolidWorks Corp., Waltham, MA, USA) that accommodated both the transducer and a load cell (Fig. 1). This design allowed for the detection of examiner forces while ensuring unimpeded transducer contact with the scanned material. The ultrasound transducer interface housing was comprised of acrylonitrile butadiene styrene (ABS) and printed using an Objet500 rapid prototyping machine (Stratasys Ltd., Eden Prairie, MN, USA). Scanning force feedback was performed using a FC22 compression load cell (Measurement Specialties, Hampton, VA, USA) for axial force measurement. This load cell features a force detection capability up to 44.5 N $\pm$ 0.5 N with non-linearity, hysteresis, and repeatability characteristics of $\pm$1%. Applied forces detected by the load cell generated signals that were sent to a laptop computer (Latitude; Dell Corp., TX, USA) through a USB port via an Arduino Uno microcontroller (Arduino LLC; www.arduino.cc). Analog signals generated by the load cell were connected to the microcontroller analog output directly without an amplifier. The microcontroller sends the force signals to the computer as a series of serial strings by using the USB port as a virtual serial port. The graphical user interface (GUI) used during the force feedback scanning was developed with C++ programing language (Microsoft, Redmond, Washington, USA) to facilitate calibration of the load cell and allow for the viewing of real-time force values during data collection. The force-feedback transducer interface system was used during all manual and automated scanning featured in this study.

Clinical sonographers have been observed (*Gilbertson & Anthony, 2013*) applying variable axial forces when performing diagnostic ultrasound examinations at the abdomen with mean values ranging from 5 N to 14 N. These high forces are often required to manipulate the relative position of superficial anatomic structures in order to obtain

optimal scans of deeper tissues. In contrast, practitioners using musculoskeletal quantitative ultrasound techniques often require minimal examiner forces which may be as low as 1 N in order to minimize tissue deformation during scanning (*Ishida & Watanabe, 2012*). Consequently, an *a priori* decision was made to use applied stress conditions during all scanning procedures in this study with target forces ranging from 1 N to 10 N, with image capture occurring at 1 N increments. The force targets were randomized for each examiner to minimize order effects on the scanning technique and the material deformation measures (VassarStats random number generator) (*Lowry, 2004*). A calibration adjustment was completed before each scan to account for the weight of the transducer, interface housing and components, and cord connecting the transducer to the ultrasound machine.

### Manual force feedback image acquisition

Manual scanning of the ultrasound phantom by the six examiners was performed using an augmented transducer for the provision of real-time force feedback.

Each examiner began scanning following the positioning of the ultrasound phantom and application of water-soluble transmission gel. The examiners operated the ultrasound machine within view of the laptop computer with the GUI featuring the real-time force control levels. Examiners attempted to exert the targeted axial force through the transducer and onto the ultrasound phantom surface without incurring any pitch or roll of the device. Image capture occurred when the GUI on the laptop computer indicated attainment of the target force ($\pm 0.5$ N), and material thickness measurements were obtained using the Sonosite ultrasound machine digital caliper measurement function. Digital caliper measures were taken at the midpoint of the region of interest, within the simulated fascial planes of the ultrasound phantom, starting from the superior fascial plane to the inferior fascial plane. Each image capture and digital caliper measure was obtained three times. Every examiner repeated this process 10 times in order to acquire the images for each target force application condition (a 1 N–10 N range using 1 N increments). The mean values obtained for the longitudinal images were used for the subsequent analyses. A total of 60 material thickness measures were acquired following the manual scanning procedures by the six examiners.

### Robot-assisted image acquisition

Robot-assisted scanning of the ultrasound phantom by a bioengineer and sonographer was performed for the sole purpose of generating reference values for the criterion-based reliability analysis.

Automated image acquisition and transducer positioning were performed with the KUKA Light Weight Robot (LWR). The KUKA LWR (Kuka Inc., Augsburg, Bavaria, Germany) has multiple joints that are equipped with position and joint torque sensors (*Archila Diaz, Suell & Noronha Castro Pinto, 2010*). Each joint of the portable robot is driven by compact brushless motors via harmonic drives, which allow for 7 degrees of freedom with an estimated motion error of $\pm 0.05$ mm. The total weight of the robot is approximately 16 kg, with a rated payload of 7 kg. The force-feedback transducer interface housing features an adapter that allowed for the external device and the transducer to be connected to the KUKA LWR end effector (Fig. 2). This feature of the design allowed

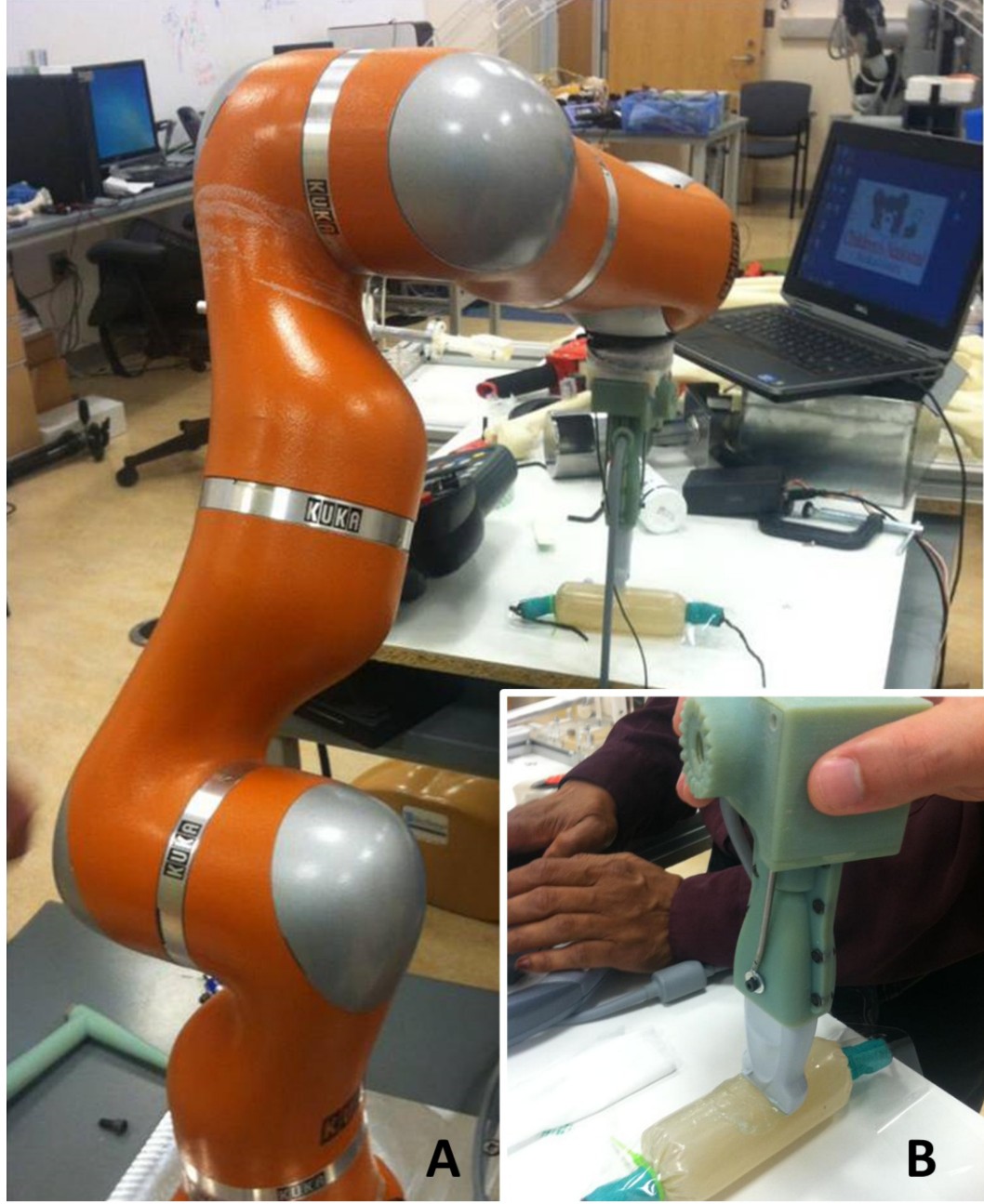

**Figure 2** **Quantitative ultrasound reliability assessed from a manual force-feedback image capture method based on criterion reference values derived from an automated image capture method.** The KUKA Light Weight Robot (A) was used to obtain automated ultrasound images for comparison with manually acquired images (B) from six examiners using force feedback. The robot-generated forces and material deformation were measured using the same ultrasound device, force transducer, and ultrasound phantom that were used by the six examiners.

for the investigators to monitor the force imposed by the robotic arm, and maintain measurement consistency between the manual and robot-assisted scanning sessions. The gravity compensation mode of the robot was used to manually place the machine in close proximity to the transducer and ultrasound phantom. The position control mode was then selected after the robot was locked into the testing location.

The robot-assisted scanning session began following the positioning of the ultrasound phantom and application of water-soluble transmission gel. Scanning involving the KUKA LWR required the bioengineer to operate the robotic arm to position the augmented transducer on the ultrasound phantom surface. Using transducer and ultrasound phantom positioning similar to the manual image acquisition sessions, the bioengineer attempted to exert the targeted axial force through the transducer onto the ultrasound phantom surface by incrementally moving the robot arm using the KUKA LWR control panel. The bioengineer ceased the movement of the robotic arm and transducer when the GUI on the laptop computer indicated attainment of the target force ($\pm$0.5 N).

The sonographer's tasks were coordinated with the efforts of the bioengineer. Once the bioengineer completed the positioning of the robotic arm and transducer, the sonographer (C.I.) initiated the image capture procedure. The sonographer operated the ultrasound machine (without manipulating the transducer) within view of the laptop computer with the GUI featuring the real-time force control levels. The sonographer confirmed the attainment of the target force ($\pm$0.5 N) per the GUI on the laptop computer, verified that the image did not contain artifacts, and then captured the image within the field of view. Following each image capture, the sonographer used the Sonosite ultrasound machine digital caliper measurement function to obtain material thickness measures. In a similar manner to the manual scanning sessions, the image capture and digital caliper measures were obtained three times. The bioengineer and sonographer repeated this process 10 times in order to acquire the images for each target force application condition (a 1 N–10 N range using 1 N increments). Following these scanning procedures, a total of 10 material thickness measures were acquired using the robot-assisted procedure to obtain the reference values used for the criterion-based reliability analysis.

## Data analysis

The interrater reliability of the examiners for the measurement of material thickness was estimated using intraclass correlation coefficients (ICC). The $ICC_{2,k}$ was used to determine the interrater reliability using a 2-way mixed model absolute agreement approach (*Portney & Watkins*, *2009*). The criteria to interpret the ICC values were based on the method provided by *Portney & Watkins* (*2009*): 00–.49 = poor reliability, .50–.74 = moderate reliability, and .75–1.00 = excellent reliability. The coefficient of variation (CV) was used to convey the estimated proportional measurement error, and the standard error of the measurement (SEM) was calculated to provide an estimate of absolute reliability of the examiners' material thickness measures. Overlay scatter plots and the coefficient of determination ($R^2$) derived from linear regression were used to convey the degree of association between material thickness measures obtained via manual and automated force-feedback image acquisition. The $R^2$ values also were used to express the agreement

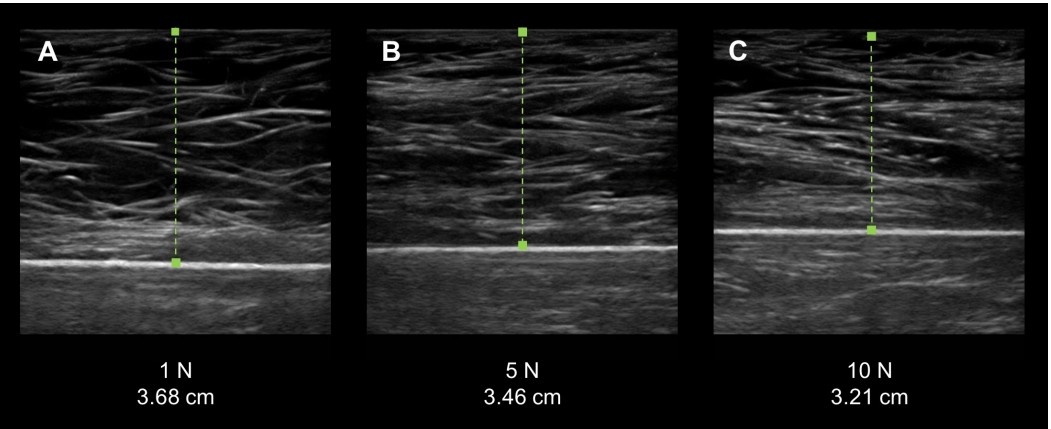

**Figure 3** **Force-feedback augmented sonography of a calibration ultrasound phantom using manual image capture methods.** (A–C) The longitudinal view exemplar ultrasound images were obtained by an examiner while using a force-feedback augmented transducer. The values below each image show the progressive increase in the targeted applied force on the ultrasound phantom and the corresponding increase in material deformation. (N, Newtons; cm, centimeters).

between the material thickness measures obtained through manual and automated means given the identical source material, transducer, and measurement approach used in both image acquisition conditions. The data featured in the linear regression includes the material thickness measures across the range of force targets, and the scale of measurement used on the ordinate and abscissa of the scatter plots are equivalent. Statistical analyses were performed using PASW Statistics for Windows, Version 18.0 (SPSS Inc., Chicago, IL, USA). All data and variance distributions were normal based on the Shapiro–Wilk and Levene's tests, and conveyed as means and standard deviations. Material thickness measures (i.e., material deformation) are expressed in cm, and force values are expressed in Newtons (N). The $\alpha$ level was set at .05, and two-tailed $p$ values <.05 are considered significant for all inferential statistics.

## RESULTS

### Measurement error from images acquired using force feedback scanning

Manual force-feedback scanning was an effective method of controlling the force exerted by the examiners onto the surface of the ultrasound phantom via the transducer. The recorded mean forces produced by the examiners matched the target forces across all intervals within an estimated −.10 N–.12 N, which did not exceed the measurement tolerance of the load cell (±0.5 N). In addition, the examiners demonstrated a low magnitude of measurement error during the manual force-feedback image capture and morphometry measurements. The ultrasound phantom exhibited increased deformation with the progressive intervals of examiner applied force with the resultant material thickness measures ranging from 3.79 cm (±.08 cm) to 3.15 cm (±0.5; Fig. 3). The proportion of measurement error observed among the examiners, as expressed with the CV, was 1.5% to 2.9%. Additionally, the absolute measurement error displayed by the examiners across the range of force targets

**Table 1 Descriptive data and measurement error estimates for the material thickness values.** The table summarizes the examiners' applied force against the ultrasound phantom surface and the corresponding material deformation.

| Applied force | | Material deformation | | |
|---|---|---|---|---|
| Target force (N) | Mean force attained (N ± SD) | Mean material thickness (cm ± SD) | CV (%) | SEM (cm) |
| 1.0 | 1.0 ± .1 | 3.79 ± .08 | 2.2 | .03 |
| 2.0 | 2.0 ± .1 | 3.58 ± .05 | 1.5 | .02 |
| 3.0 | 3.0 ± .0 | 3.53 ± .10 | 2.9 | .04 |
| 4.0 | 4.0 ± .1 | 3.41 ± .08 | 2.5 | .03 |
| 5.0 | 5.0 ± .1 | 3.35 ± .06 | 1.9 | .02 |
| 6.0 | 6.1 ± .1 | 3.34 ± .06 | 1.9 | .02 |
| 7.0 | 7.1 ± .0 | 3.27 ± .07 | 2.1 | .02 |
| 8.0 | 8.1 ± .0 | 3.23 ± .07 | 2.2 | .02 |
| 9.0 | 9.0 ± .1 | 3.20 ± .08 | 2.4 | .03 |
| 10.0 | 10.1 ± .1 | 3.15 ± .05 | 1.6 | .02 |

**Notes.**
N, Newtons; SD, standard deviation; cm, centimeters; CV, coefficient of variation; SEM, standard error of the measurement.

**Table 2 Interrater reliability among all examiners using manual image capture with force feedback.** Intraclass correlation coefficient ($ICC_{2,k}$) for averaged material thickness measures across all examiners.

| | $ICC_{2,k}$ | F Test | | | |
|---|---|---|---|---|---|
| | | Value | df1 | df2 | p-value |
| All examiners | .98 | 97.02 | 9 | 45 | <.001 |

**Notes.**
df, degrees of freedom.

based on the SEM was .02 cm–.04 cm. The mean force application and material thickness values for the examiners are provided in Table 1.

## Examiner interrater reliability across examiners and relative to the criterion-based reference measurements

Interrater reliability for the group of examiners was excellent for the material thickness measures obtained using the manual force feedback image capture method (Table 2). The high degree of measurement consistency among the examiners was reflected by the $ICC_{2,k}$ value of .98 ($p < .001$). Moreover, individual examiner performance in comparison with criterion-based reference measures obtained from the automated image capture procedure was also excellent. No examiner within the three sonography experience categories exhibited an $ICC_{2,k}$ value lower than .91 ($p < .001$). The interrater reliability estimates for each examiner using the manual force image feedback image capture method in comparison to the automated capture method are summarized in Table 3. The degree of positive association between both methods of image capture was large with the $R^2$ values ranging from .86 to .98 ($p < .001$). One of the two novice examiners had a $R^2$ value below .90 regarding the association between material thickness measures obtained with manual force feedback image capture and the criterion-based measures obtained

**Table 3 Criterion-based reliability for each examiner using manual image capture with force feedback.** The table features estimates of criterion-based reliability. The estimates were calculated using the individual examiner material thickness measures obtained with manual force-feedback scanning in comparison with reference measures obtained from the automated scanning method with the KUKA Light Weight Robot. The intraclass correlation coefficient ($ICC_{2,k}$) values are based on the averaged material thickness measures obtained by each of the examiners.

| Examiner experience level | | $ICC_{2,k}$ | F test | | | |
|---|---|---|---|---|---|---|
| | | | Value | df1 | df2 | p-value |
| *Experienced (>10 years)* | Examiner 1 | .98 | 54.07 | 9 | 9 | <.001 |
| | Examiner 2 | .92 | 37.95 | 9 | 9 | <.001 |
| *Intermediate (1 year)* | Examiner 3 | .91 | 120.53 | 9 | 9 | <.001 |
| | Examiner 4 | .97 | 101.64 | 9 | 9 | <.001 |
| *Novice (1 month)* | Examiner 5 | .92 | 22.84 | 9 | 9 | <.001 |
| | Examiner 6 | .99 | 109.84 | 9 | 9 | <.001 |

**Notes.**
df, degrees of freedom.

with automated image capture. Nevertheless, the examiners generally exhibited excellent interrater reliability for measures of material thickness over a range of stress conditions, and significant correspondence with the criterion-based measures. The overlay scatter plots and $R^2$ values depicting the relationship between the criterion-based measures and each of the examiners' measures are provided in Fig. 4.

## DISCUSSION

We examined the interrater reliability of force-feedback scanning to acquire material thickness measures over a range of examiner-generated applied stress conditions. All of the examiners demonstrated excellent interrater reliability across all of the force targets while scanning the custom muscle tissue-mimicking ultrasound phantom. The examiners also appeared to attain reliable material thickness measures relative to the criterion-based measures obtained with the use of automated image capture and stress application methods during scanning.

Although it has been suggested that variations in examiner performance may adversely affect measures of morphometry and morphology (*Pillen & Van Alfen*, *2011*; *Ishida & Watanabe*, *2012*; *Wagner*, *2013*), good evidence exists that supports the reliability of selected quantitative ultrasound techniques without augmented feedback. Acceptable intrarater reliability for diagnostic ultrasound assessment has been found for tests involving the thickness and cross-sectional area of the rectus femoris (*Bemben*, *2002*) ($ICC_{3,2} = .72-.99$, $p < .05$; $CV = 3.5\%-6.7\%$) and similar morphometry measures for the trapezius (*O'Sullivan et al.*, *2007*) have also been reported as reliable ($ICC_{3,3} = .88-.96$, $p < .05$). In addition, acceptable levels of examiner performance have been reported for the between-day, interrater reliability of multifidus (*Sions et al.*, *2014*) thickness measures (L4-5) in older adults ($ICC_{3,3} = .86$, $p < .05$; mean thickness = 3.19 cm $\pm$ .73 cm, SEM = .29 cm). Clearly, examiners in research settings have demonstrated acceptable reliability for muscle thickness measures and basic assessments of muscle morphology (*Zaidman et al.*, *2014*). However,

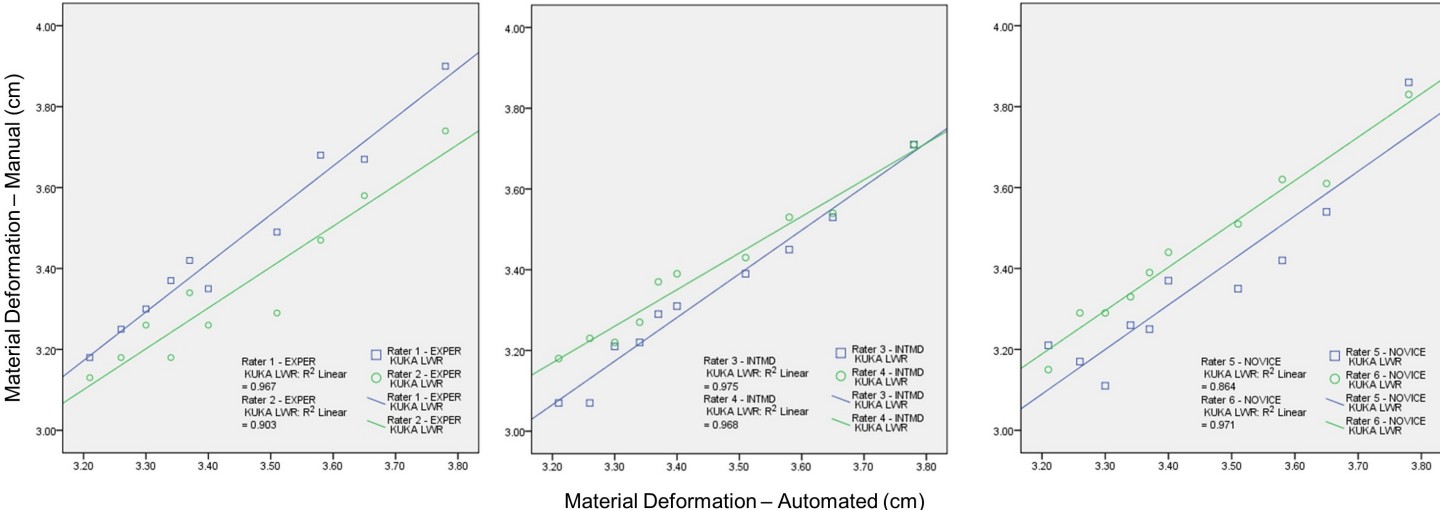

**Figure 4  The association between material thickness measures obtained via the manual force-feedback image capture method and automated image capture method.** The KUKA Light Weight Robot (LWR) was used to obtain automated ultrasound images for comparison with manually acquired images from six examiners using force feedback. Deformation of the phantom material, secondary to progressive intervals of applied manual or automated force, was measured during the scanning procedures. Therefore, higher values along the ordinate and abscissa are associated with lower stress levels. These procedures were conducted using the same ultrasound machine and transducer, the force-feedback interface system, and muscle tissue-mimicking ultrasound phantom for both image acquisition methods. The overlay scatter plots depict the material thickness measures obtained with automated image acquisition along the abscissa, and the corresponding values for material thickness obtained with manual force feedback image acquisition along the ordinate. The coefficient of determination ($R^2$) between each examiner and the KUKA LWR depicts a significant association among the serial material thickness measures attained by each of the examiners (varying from experienced to novice) with those attained using the automated image capture method over a range of force targets (1 N–10 N in 1 N increments; $R^2 = .86$–$.97$, $p < .001$; Experienced (EXPER), >10 years; Intermediate (INTMD), 1 year; Novice, 1 month; N, Newtons; cm, centimeters).

additional study is needed to determine whether a variety of quantitative ultrasound techniques may be reliably performed in typical clinical environments. Moreover, the range of quantitative ultrasound techniques vary from muscle thickness measures to identifying areas of hyperechoic tissue for biopsy site identification (*Pillen et al.*, *2007*). The latter technique may involve alteration of both the transducer orientation and the force exerted by the examiner. Also, sonographers may apply high transient peak forces during clinical ultrasound examinations, which may yield serial images with different levels of tissue deformation (*Gilbertson & Anthony*, *2013*). Consequently, feedback enhanced sonography may become an important component of the continual development of diagnostic ultrasound applications and capabilities.

Other investigators have used enhanced diagnostic ultrasound to aid the assessment of musculoskeletal structures. Early efforts included the use of M-mode ultrasound scanning with a load cell interface to determine tissue characteristics during palpation (*Zheng & Mak*, *1996*; *Zheng et al.*, *2006*). Tissue thickness and elasticity have been calculated for the transverse carpal ligament using linear interpolation analysis of critical points derived from ultrasound echoes (*Zheng et al.*, *2006*). These ultrasound echo critical points were measured using a series of applied stress conditions with a peak force load of 20 N. *Burcher et al.* (*2005*) designed an ultrasound interface system with a load cell and an optical localizer for B-mode freehand scanning. However, their approach was to detect examiner force and transducer

position in order to create tissue deformation corrected B-mode scans using an elastic model specific to a given individual or material. The investigators were able to validate their approach to B-mode scanning image correction with criterion-based measures obtained from a gelatin-based ultrasound phantom and a materials testing machine, demonstrating excellent agreement for both force application and material deformation values (*Burcher et al.*, *2005*). The measurement approach used by *Chadli et al.* (*2012*) and *Gilbertson & Anthony* (*2013*) reflects the general image acquisition and measurement strategy employed in this report. These investigators utilized real-time force feedback via a computer GUI to inform the examiner of instantaneous axial forces exerted against the scanning surface. *Chadli et al.* (*2012*) developed their force and position sensing interface for potential telehealth and robot-assisted sonography procedures, and *Gilbertson & Anthony* (*2013*) have previously used their force/torque measuring system to address the ergonomic concerns of clinical sonographers and to characterize the amount of examiner-generated stress during examinations. In contrast, this study focuses on how knowledge of performance via force feedback affects scanning consistency at targeted force levels encountered in a variety of quantitative ultrasound procedures. While the real-time visual feedback from the ultrasound monitor may aid the performance of a quantitative ultrasound examination, this form of feedback is insufficient to inform the examiner how to adjust applied forces in order to maintain similar levels of tissue strain within or between scanning sessions.

The examiners in this study were categorized based on their background in sonography: experienced (>10 years), intermediate (1 year), and novice (1 month). The interrater reliability attained by each examiner in relation to the criterion-based measurement values suggests that force-feedback scanning performance is not dependent on the duration of sonography experience. The lowest estimates of interrater reliability, a reliability coefficient of .91 or .92 ($ICC_{2,k}$), were attained by one examiner in each sonography experience level category (i.e., Examiner #2, #3, and #5; Table 3). Regarding the degree of association and agreement between the measures obtained from the manual and automated image capture, the coefficient of determination ($R^2$) was above .90 for all examiners with the exception of one novice examiner (Examiner #5; $R^2 = .86$). While the overlay scatter plots depict strong association between each examiner and the criterion-based measures, differences among the examiners paired by experience level did emerge. The intermediate examiners appeared to have a departure in measurement consistency as the applied forces increased, whereas the experienced examiners displayed their greatest measurement consistency at the highest levels of applied force (Fig. 4). However, the mean measurement difference among the examiners paired by experience was modest and ranged between .04 cm and .12 cm. Moreover, the magnitude of the measurement errors does not appear to be systematic based on the CV and SEM values from the collective performance of the six examiners across all force target levels (Table 1). Based on our early findings, force feedback scanning may provide an effective means of obtaining consistent quantitative ultrasound values among examiners with varied experience levels. Nevertheless, it is important to note that the differences in material deformation values obtained by the subgroups were not subject to statistical analysis in this study. Additional investigative work is needed in the applied use of enhanced-sonography in clinical settings.

One limitation of this work is that the operational definition for sonography experience was based only on the duration of time between initial training and the start of data collection. However, given the differences in psychomotor skills associated with quantitative ultrasound in comparison to other forms of clinical ultrasound, experienced clinical sonographers often have to adapt to the relatively low forces associated with material thickness measures. The experienced examiners in this study included an investigator with training exclusively with quantitative ultrasound methods (Examiner #1), and a longtime professional sonographer with less experience in quantitative musculoskeletal measurements (Examiner #2). Insights shared by the participating professional sonographer were consistent with the observation by Smith-Guérin et al. (2003) that clinical sonography procedures may feature applied forces generally within 5 N–20 N. Also, a potential study design constraint was that the measurement performance of the examiners was only assessed while using the force-feedback transducer interface system during manual scanning, so no comparison could be made with unaugmented scanning. However, it is not possible for multiple examiners to attain similar scanning force levels, across a range of force targets, without a form of augmented scanning that provides objective feedback concerning imposed forces or material deformation. Indeed, the findings in this report suggest that it is feasible for multiple examiners to measure material thickness reliably at specified levels of applied force when real-time feedback is provided regarding the magnitude of force imposed on the ultrasound phantom surface. Nevertheless, there may be some value in further study to examine manual scanning with and without force feedback for quantitative procedures restricted to very low examiner forces. Furthermore, follow up investigations regarding the applied use of feedback-enhanced sonography should include morphology measures from individuals with musculoskeletal or neuromuscular disorders. This approach to ultrasound scanning and image capture may have utility for clinical assessment, or training investigators and practitioners who are new to quantitative ultrasound methods.

The findings of this study were also limited by the constraints associated with the force-feedback transducer interface prototype. This device was a second-generation unit with single-axis force detection capability, which differs from more advanced devices with multi-axis force/torque detection and transducer orientation feedback (Chadli et al., 2012; Gilbertson & Anthony, 2013). A previous investigation concerning manual force-feedback scanning using a curvilinear transducer and a six-axis force/torque measuring system revealed that the largest proportion of detected forces were exerted through the main axis of the transducer (i.e., the $y$-axis) (Gilbertson & Anthony, 2013). The initial design and instrumentation of the prototype featured in this study does not account for sheer forces generated during typical musculoskeletal scanning or the torque generated by rolling the transducer (along the $x$-axis) on the surface of the scanned material. While these ancillary motions give rise to force levels that may exceed load cell measurement error, they are less pronounced during the quantitative measurement of material thickness using a linear transducer as featured in this study. Consequently, the forces detected by the ultrasound interface prototype used in this study likely reflect the most critical axial stresses affecting the material deformation during the quantitative ultrasound scanning procedure. Finally, the automated image capture method using the KUKA LWR was solely incorporated to

obtain criterion-reference values used as a comparator to the image capture and material thickness measures obtained by the examiners using force-feedback augmented manual scanning. Although the automated image capture method was facilitated by the use of the KUKA LWR to control the transducer position and force, the measurement of the resultant material deformation was measured in a conventional manner by an examiner. While this aspect of the measuring procedure introduces an additional source of error into the criterion-referenced scans, the robot-assisted method circumvents the psychomotor element associated with handling the transducer which constitutes the largest source of error in quantitative ultrasound imaging (*Chadli et al.*, *2012*; *Gilbertson & Anthony*, *2013*; *Harris-Love et al.*, *2014*). Therefore, the criterion-reference measures obtained via automated scanning allowed for us to more fully characterize the reliability performance of the examiners during their use of force-feedback augmented manual scanning.

In conclusion, force feedback enhanced manual ultrasound scanning allows for the reliable acquisition of material thickness measures over a range of examiner-generated applied force conditions. The reliability of this imaging method is excellent among a group of examiners with varied diagnostic ultrasound experience. Moreover, the quantitative measures of the custom muscle mimicking ultrasound phantom by the examiners are positively associated with criterion-based values obtained through automated scanning. The relatively low magnitude of error associated with the examiners' performance merits the continued development of this approach to diagnostic imaging in clinical and research settings. Future development efforts should also address if enhanced sonography, via the production of force feedback accessory units or transducers with integrated sensors, can be adequately scaled to be viable within the clinical practice environment. This process would involve addressing the issues of cost and access, compatibility of the technology across an array of device models, and human factors design to ensure effective usage by practitioners.

## ACKNOWLEDGEMENTS

The authors thank the staff of the Clinical Learning and Simulation Skills (CLASS) Center at The George Washington University for allowing our group to use the ultrasound machine, and Dr. Brian Garra at the Washington DC Veterans Affairs Medical Center for provision of the ultrasound phantom. In addition, we thank Dr. Kevin Cleary of the Sheikh Zayed Institute for Pediatric Surgical Innovation at Children's National Hospital in Washington, D.C. for providing access to the KUKA Light Weight Robot. Any opinions or recommendations expressed in this publication are those of the authors and do not necessarily reflect the view of the US. Department of Veterans Affairs or the National Center for Advancing Translational Sciences or the National Institutes of Health.

### Funding

This publication was partially supported by Award Number UL1TR000075 and UL1TR000101 from the NIH National Center for Advancing Translational Sciences

(NCATS), National Institutes of Health (NIH), through the Clinical and Translational Science Awards Program (CTSA), and a VISN 5 Pilot Research Grant (VISN 5; VA Station: 688)—VHA/VA Capitol Health Care Network. The funders had no role in study design, data collection and analysis, decision to publish, or preparation of the manuscript.

### Grant Disclosures

The following grant information was disclosed by the authors:
NIH National Center for Advancing Translational Sciences (NCATS): UL1TR000075, UL1TR000101.
National Institutes of Health (NIH).

### Competing Interests

The authors declare there are no competing interests.

### Author Contributions

- Michael O. Harris-Love conceived and designed the experiments, performed the experiments, analyzed the data, contributed reagents/materials/analysis tools, wrote the paper, prepared figures and/or tables, reviewed drafts of the paper.
- Catheeja Ismail and Reza Monfaredi conceived and designed the experiments, performed the experiments, contributed reagents/materials/analysis tools, wrote the paper, reviewed drafts of the paper.
- Haniel J. Hernandez performed the experiments, wrote the paper, reviewed drafts of the paper.
- Donte Pennington wrote the paper, reviewed drafts of the paper.
- Paula Woletz, Valerie McIntosh and Bernadette Adams performed the experiments, reviewed drafts of the paper.
- Marc R. Blackman analyzed the data, reviewed drafts of the paper.

### Ethics

The following information was supplied relating to ethical approvals (i.e., approving body and any reference numbers):
The Washington DC VAMC Institutional Review Board (IRB; #01671) and Research and Development Committee.

### Data Availability

The data is owned by the US Department of Veterans Affairs and is therefore restricted.

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
