# Peer review of "Interrater reliability of quantitative ultrasound using force feedback among examiners with varied levels of experience"

_PeerJ, doi:10.7717/peerj.2146_

## Round 0.1 · original submission · Minor Revisions

The reviewers are impressed with many aspects of the manuscript and study. Additonal clarifiication is required on some aspects of the image acquisition process and how this was actually performed. Such details are vital if this method is to be used in clincial practice.

·

Basic reporting

No mention of funding. It seems like a study which may not have been possible without associated funding (robotics etc). If this is the case, then the authors should be commended. However this might be worth double checking.
As a reviewer I lack the technical knowledge of the processes behind the development of the device and processes. However my comments will be designed around the practical and clinical applications of the paper.
ABSTRACT:
This section is concise and contains the required information.
TITLE:
The title is appropriate
INTRODUCTION:
The introduction is well structured and provides sufficient direction to the development of the hypothesis and aims.

Experimental design

The number of sonographers may be too minimal to draw significant causational conclusions from. This may explain the lack of difference between the levels of expertise in the study. As this wasn’t a main aim of the study, the paper should not be rejected because of it. However I believe it does limit some of the implications to be drawn from only 6 sonographers (2 per level of expertise). The authors should consider mentioning this point in their limitations section. I was quite surprised there was no difference in the results between the levels of proficiency.

Validity of the findings

No Comments

Additional comments

As a reviewer I lack the technical knowledge of the processes behind the development of the device and processes. However my comments will be designed around the practical and clinical applications of the paper.

The authors should be commended for their work. It is a well thought out study with some useful practical implications. However I believe determining how to implement these findings in a practical setting is the next step. I understand the paper current paper was more of a ‘proof of concept’ approach and it achieves this. However I believe it might be worth spending sometime on trying to figure out how this might have some transmission to practice. Are we all going to purchase these housings and use them when scanning? From a clinical perspective, it doesn’t seem to have the necessary links to its application. Yet from a research and laboratory perspective, it is golden! It is a great way of standardizing process of data acquisition.

·

Basic reporting

In general this is a well reported reliability study. The premise for the study is sufficient. The methods suitable and the results are reported adequately. Finally, the authors critically analyse their data well, and put the results into context with other similar approaches that have been utilised. My main criticism that I believe need editing is that the section explaining the automated image acquisition needs to be improved. It took me three reads to know what this process was about, so I suggest that the paragraph of 165-186 should be prefaced with a sentence explaining what this process is required for (ie. to create criterion referenced measures for comparison).

Experimental design

The experimental design is certainly adequate and the aims clear and approach to achieve the aims suitable. My only criticism of the reporting of the methods is that it is not mentioned anywhere how the images were collected and whether this was a manual or automated process. Were the images manually recorded when the target forces were met, or was this process automated? Some extra detail is needed here, because this could potential influence the results.

Validity of the findings

The data is robust and the statistics are suitable and the conclusions are reasonable - the reliability is excellent.

---

## Round 0.2 · accepted · Accept

We appreciate the effort you have made in addressing some of the initial concerns of the reviewers. As a consequence we are happy to recommend acceptance of this manuscript.

·

Basic reporting

No comments

Experimental design

No comments

Validity of the findings

No comments

Additional comments

All revisions address the concerns raised.

·

Basic reporting

I am happy with the changes made to the document. The description of how the images were acquired and the changes to the aims of the study make things more clear.

Experimental design

I believe this is appropriate for the purposes of the analysis.

Validity of the findings

The findings are robust, statistically sound and controlled.